# Identification of Factors Linked to Higher Water-Deficit Stress Tolerance in *Amaranthus hypochondriacus* Compared to Other Grain Amaranths and *A. hybridus*, Their Shared Ancestor

**DOI:** 10.3390/plants8070239

**Published:** 2019-07-22

**Authors:** Tzitziki González-Rodríguez, Ismael Cisneros-Hernández, Jonathan Acosta Bayona, Enrique Ramírez-Chavez, Norma Martínez-Gallardo, Erika Mellado-Mojica, Mercedes G. López-Pérez, Jorge Molina-Torres, John Délano-Frier

**Affiliations:** Centro de Investigación y de Estudios Avanzados del I. P. N., Unidad Irapuato, Km 9.6 del Libramiento Norte Carretera Irapuato-León, C.P. 36821 Irapuato, Guanajuato, Mexico

**Keywords:** Abscisic acid, grain amaranth, osmotic adjustment, sucrolytic enzymes, sugar starvation response, water-deficit stress tolerance

## Abstract

Water deficit stress (WDS)-tolerance in grain amaranths (*Amaranthus hypochondriacus*, *A. cruentus* and A. *caudatus*), and *A. hybridus*, their presumed shared ancestor, was examined. *A. hypochondriacus* was the most WDS-tolerant species, a trait that correlated with an enhanced osmotic adjustment (OA), a stronger expression of abscisic acid (ABA) marker genes and a more robust sugar starvation response (SSR). Superior OA was supported by higher basal hexose (Hex) levels and high Hex/sucrose (Suc) ratios in *A. hypochondriacus* roots, which were further increased during WDS. This coincided with increased invertase, amylase and sucrose synthase activities and a strong depletion of the starch reserves in leaves and roots. The OA was complemented by the higher accumulation of proline, raffinose, and other probable raffinose-family oligosaccharides of unknown structure in leaves and/or roots. The latter coincided with a stronger expression of *Galactinol synthase 1* and *Raffinose synthase* in leaves. Increased SnRK1 activity and expression levels of the class II *AhTPS9* and *AhTPS11* trehalose phosphate synthase genes, recognized as part of the SSR network in Arabidopsis, were induced in roots of stressed *A. hypochondriacus*. It is concluded that these physiological modifications improved WDS in *A. hypochondriacus* by raising its water use efficiency.

## 1. Introduction

Plants have evolved to avoid, escape or tolerate water deficit stress (WDS) conditions in order to minimize the negative effects it has on plant growth, survival, yield and geographical distribution. The controlling mechanisms for plant performance under drought are complex, due to the multifaceted interplay between transcription factors (TFs), protein kinases and phosphatases, microRNAs, hormones, several types of stress-related proteins, co-factors, ions, and metabolites. Their effect can also be manifold, being manifested at molecular, cellular, metabolic, morphological, physiological and/or developmental levels [1,2,3]. Acting in combination, they increase the possibility of survival under moderate to severe WDS. TFs are involved in plant adaptation to drought, acting either as stress sensors or as moderators of stress-responsive gene expression, which depend primarily on the accumulation of abscisic acid (ABA), a key plant stress-signaling hormone. ABA-dependent pathways are controlled by AREB/ABFs TFs, whereas ABA-independent signaling pathways are regulated mostly by DREB2A TFs [4]. However, a dynamic cross-talk between ABA-dependent and ABA-independent signaling pathways, regulated by Ca^2+^, jasmonic acid, ethylene, gibberellins, auxins, reactive oxygen species (ROS), G proteins and other factors, is usually established in water-stressed plants. Among the numerous responses activated to mitigate drought stress are: (i) Stomatal closure to prevent water loss through transpiration; (ii) maintenance of photosynthetic activity under stress or reversible inhibition of photosynthesis; (iii) activation of mitochondrial alternative oxidase and antioxidant systems needed for the removal of deleterious ROS accumulation; (iv) altered source-sink relations and carbon (C) partitioning; (v) changes in root architecture and/or induction of root growth via the inhibition of ethylene accumulation; (vi) osmolyte synthesis accumulation (e.g., soluble non-structural carbohydrates [NSCs], raffinose-family oligosaccharides [RFOs] and/or proline [Pro] and others) for osmotic adjustment (OA); (vii) regulation of aquaporin and ion channel activity; and (viii) altered expression of genes coding for several other TFs, heat shock proteins, late embryogenesis abundant proteins (LEAs), NSC cleaving enzymes and/or enzymes required for the biosynthesis of osmoprotectants and antioxidants, among others. Moreover, epigenetic regulation and RNA-related processes, in addition to post-translational regulation by phosphorylation, ubiquitination and sumoylation are also involved [5,6,7,8].

The genus *Amaranthus* consists of 60–70 species. Some are consumed as vegetables or are used as a source of grain. The latter (i.e., *Amaranthus hypochondriacus*, *A. cruentus*, and *A. caudatus*) possess desirable agronomic characteristics and produce highly nutritional seeds. Moreover, they can adapt to drought and poor soils [9]. Domesticated grain amaranths presumably descend from wild *A. hybridus*, although their origin and taxonomic relationships are still uncertain [10,11]. Compared to vegetable amaranths [12,13], only a handful of studies have been performed to determine the nature of the traits that enable grain amaranths to thrive in harsh conditions, such as drought stress [14,15]. Thus, the understanding of the drought tolerance mechanism(s) that operate in grain amaranth remains limited. Natural variations in WDS tolerance are common in plants, as established by copious information describing differences between natural accessions, wild populations and domesticated C3 plants, including rice [16,17,18], C3 and C4 grasses [19], different C4 subtypes [20,21], or contrasting genotypes of commercially important C3 and C4 crops [5,22,23]. Based on this evidence, and considering their different geographic origin and preferential adaptation to temperate or tropical conditions, it was hypothesized that WDS tolerance in grain amaranths and *A. hybridus*, their shared ancestor, would vary significantly between species. To validate this hypothesis, these four amaranth species were subjected to a battery of tests to identify variations in responses strongly associated with increased WDS tolerance, such as osmotic adjustment (OA), ABA-dependent signaling and the implementation of a sugar starvation response (SSR). Therefore, assays were performed, in leaves and roots, to monitor modified gene expression of ABA-related genes, RFO-biosynthetic genes, trehalose (Tre) biosynthesis and degradation genes, including class I and II trehalose phosphate synthase genes and other genes associated with the SSR network. The content of RFOs, soluble and insoluble NSCs, Tre and Pro, as well as changes in invertase, sucrose synthase and amylase activities, was also determined to analyze differences in OA. Additionally, SnRK1 and TOR kinase activities were assayed, considering their opposing roles in the control of metabolism and growth in response to fluctuating sugar levels that characterize the SSR [24]. The results obtained suggest that tolerance to WDS in grain amaranths is multifactorial, and that the tolerance range observed between species involves the differential activation of distinct WDS-amelioration mechanisms. These findings are discussed in the context of the various strategies for WDS tolerance known to be employed by plants.

## 2. Results

### 2.1. WDS Tolerance Varies among Amaranth Species 

The initial screening to detect differences in WDS tolerance between amaranth species revealed that *A. hypochondriaucs* cv. Gabriela, was the most WDS tolerant species (with average leaf water potentials [ψ_Lav_] of −0.23 and −2.80 MPa and recovery [R] rates of 90% and 30% after withholding watering for 5 and 10 days, respectively). *A. cruentus* cv. Amaranteca was intermediate (ψ_Lav_ of −0.27 and −4.05 MPa and R rates of 60% and 30%), while *A. caudatus* was the most sensitive (ψ_Lav_ of −0.41 and −8.25 MPa and R rates of 70% and 0%). Not unlike resurrection plants, *A. hybridus* plants became severely desiccated under both treatments, to the extent that no handling was possible due to the fragility of the leaves. However, they were able to recover, to a certain extent, several days after watering was restored (Appendix A). 

### 2.2. WDS Tolerance Is Associated with Changes in NSCs Content Coupled to Sucrolytic and Amylolytic Activity in Leaves and Roots

Changes in carbohydrate concentration can contribute importantly to the OA required to maintain leaf turgor and improve soil water extraction under reduced water potentials [25,26]. Soluble NSCs showed a clear tendency to increase concomitantly with WDS intensity and to decline during recovery (R) in leaves of all species examined, except in *A. hypochondriacus*, where hexose (Hex) levels were drastically reduced under severe water-deficit stress (SWDS) and slightly increased under R (Figure 1A,B). In roots, the soluble NSC content variation in *A. hypochondriacus* differed from the rest (Figure 2). Here glucose (Glu) and fructose (Fru) levels were the highest detected, while sucrose (Suc) was the lowest. Another difference observed was that *A. hypochondriacus* and *A. cruentus* underwent a sharper depletion of starch in response to WDS in leaves and roots, particularly during SWDS. Starch levels were also significantly lower in WDS-tolerant species after R (Figure 1D and Figure 2D).

WDS stress did not increase invertase activity in leaves of *A. hypochondriacus* plants, contrary to the other species tested, which showed significant changes in foliar CWI and CI. No changes in VI activity were observed in leaves of stressed plants, irrespective of species (Figure 3). No relationship was observed between Hex and Suc levels and invertase activity in leaves of stressed amaranth plants (Figure 1 and Figure 3).

Conversely, the higher Hex/Suc ratio observed in roots of stressed *A. hypochondriacus* plants, as inferred from data shown in Figure 2A–C, was consistent with an increased CWI and VI activity under moderate water-deficit stress (MWDS) and SWDS (Figure 4A,B). In *A. caudatus* roots, the significantly reduced CWI, VI and CI activities under MWDS and SWDS were consequent with lower Hex/Suc ratios (Figure 2A–C and Figure 4).

The high constitutive sucrose synthase (SuSy) activity levels observed in roots of *A. hypochondriacus*, which underwent a further increase (ca. 2-fold) under SWDS, probably contributed to maintaining the high, stress-related, Hex/Suc in this organ, as well (Figure 5). SuSy activity was also induced under SWDS in roots of *A. caudatus*, but at levels that were ca. 8-fold lower than those detected in *A. hypochondriacus* (Figure 5). No SuSy activity was detected in leaves of the amaranth species tested, irrespective of the treatment applied (results not shown). Amylolytic activity reached the highest levels in leaves of plants subjected to SWDS, regardless of species. However, its induction in MWDS conditions was observed only in WDS-sensitive species (Figure 6A). In roots, an increase in amylolytic activity under both WDS and R conditions occurred only in *A. hypochondriacus* (Figure 6B).

### 2.3. WDS Tolerance and Changes in RFO Accumulation in Leaves and Roots

The RFO accumulation pattern in leaves and roots (Figure 7 and Figure 8) partially coincided with the expression of RFO biosynthesis-related genes (see below). Higher raffinose (Raf) accumulation in response to WDS was detected in leaves of *A. hypochondriacus* and *A. cruentus*, whereas this RFO content declined during R in all species tested (Figure 7C). Verbascose (Ver) amount increased more abundantly in *A. caudatus*, and *A. hybridus* plants (Figure 7E) and myo-inositol (MI) accumulated significantly in response to SWDS in leaves of all amaranth species analyzed, except *A. hypochondriacus* (Figure 7A).

Increased accumulation of galactinol (Gol) (in MWDS and SWDS) and Raf (in MWDS) was detected in stressed *A. hypochondriacus* roots (Figure 8B,C). Staquiose (Sta) and Ver accumulated to similar levels in response to SWDS in roots of all species analyzed (Figure 8D,E), whereas MI accumulation was triggered by SWDS and/or R in roots of *A. cruentus* and *A. hypochondriacus* plants, respectively (Figure 8A). In addition, several unknown compounds having longer retention times, suggestive of RFOs with higher degrees of polymerization, were detected (Appendix A). At least three of these putative RFOs (with retention times of 16.1, 22.2 and 33.8 min, respectively), were most abundant in leaves of *A. hypochondriacus* and could, therefore, be considered as contributors to WDS tolerance in this species.

The expression of amaranth genes coding for enzymes involved in RFOs biosynthesis (i.e., Gol synthases [*AhGolS1* and *AhGolS2*], Raf synthase [*AhRafS*], and Sta synthase [*AhStaS*]) was also analyzed. All genes were named according to the closest homology shown with their respective *Arabidopsis thaliana* orthologs. In general, no obvious relation between RFO-related gene expression (Figure 9) and RFOs content detected in response to WDS and/or R was observed.

WDS almost universally induced *AhGolS1* and *AhRafS* in amaranth leaves, whose expression tended to be highest in *A. hypochondriacus* during SWDS (Figure 9A). Moreover, WDS-induced *AhGolS1* and *AhRafS* expression levels coincided with the significantly higher Raf accumulation detected in leaves of *A. hypochondriacus* and *A. cruentus*. Foliar expression of *AhGolS2* and *AhStaS* was mostly unaffected by WDS. *AhRafS* and *AhStaS* were more strongly expressed the in roots of *A. hypochondriacus* and *A. cruentus* (Figure 9B). 

### 2.4. WDS Tolerance and Changes in Pro and Tre Accumulation in Leaves and Roots

Pro accumulation has been found to promote plant recovery from drought stress via several mechanisms. Accordingly, Pro levels were significantly higher in leaves of WDS-tolerant species. Importantly, a ca. 3-fold difference in Pro leaf content was observed between *A. hypochondriacus* and *A. caudatus* leaves under MWDS and SWDS (Figure 10A). In roots, a significant Pro accumulation was detected, mostly during SWDS, in all amaranth species tested (Figure 10B). MWDS and R induced a lower, but still significant, Pro accumulation in roots of all species analyzed, except in *A. caudatus*.

Tre is known to be present in trace amounts in most plants. It was analyzed considering its presumed involvement in the regulation of plant abiotic stress resistance [27]. The results shown in Figure 11 indicate that a 2-to-3-fold Tre accumulation was induced by both MWDS, SWDS, and sometimes in R, in both leaves (Figure 11A) and roots (Figure 11B) of the four amaranth species tested. However, the effect was stronger in the roots of WDS-susceptible species. 

Tre accumulation showed a poor correlation with the expression of genes coding for the enzymes required for Tre biosynthesis and degradation. Treahalose-6-phosphate (T6P), also considered to be a central regulator of metabolism by acting as a sensor of Suc levels [27], is synthesized from UDP-glucose and Glu by T6P synthase(s) (TPSs). Tre generation proceeds next via de-phosphorylation of T6P by T6P-phosphatases (TPPs), followed by cleavage into two Glu moieties by trehalase (TRE). This analysis was justified by previous in silico *and* in vitro data showing a relationship between many trehalose biosynthesis genes and ABA, drought, salt, and cold stress in plants [29,30]. The Tre-related grain amaranth genes examined in this study were *AhTPS1* and all class II *TPS* genes (*AhTPS5*, *AhTPS7*, *AhTPS8*, *AhTPS9*, *AhTPS10* and *AhTPS11*) annotated in the *A. hypochondriacus* genome [31]. In addition, three *TPP* genes (*AhTPPA*, *AhTPPD* and *AhTPPI*) and the only *TRE* gene identified (*AhTRE*) were included. All genes were named according to the closest homology shown with their respective *A. thaliana* orthologs (Appendix A). Within class I TPSs (i.e., *AtTPS1–AtTPS4*), only *AtTPS1,* and perhaps *AhTPS4* [32], are known to synthesize T6P, whereas class II proteins (*AtTPS5–AtTPS11*) have no such activity. However, they may possess a regulatory function, possibly associated with T6P [29]. The expression of the class I *AhTPS1* was unaffected by WDS and R in leaves of WDS-tolerant amaranths. In WDS-sensitive amaranths, this gene was induced under R (Figure 12A). The foliar expression of class II *AhTPS5*, *AhTPS7* and *AhTPS8* was mostly unaffected or downregulated by stress in WDS-tolerant amaranths. In particular, *AhTPS5* was downregulated in *A. hypochondriacus* under all conditions tested. In contrast, *AhTPS9*, but mostly *AhTPS11*, responded strongly to SWDS in all plants analyzed. The latter gene was similarly induced under MWDS and *R. AhTPPs* and *AhTRE* were generally unaffected by WDS (Figure 12A). In *A. hypochondriacus*, however, WDS conditions downregulated *AhTPPD and AhTPPI*, whereas *AhTRE*, while *AhTPPA and AhTPPD* were induced under R.

In roots, the effect that WDS and/or R had on the expression of these genes was weaker (Figure 12B). Notable changes were the downregulated expression of *AhTPS1* in *A. hypochondriacus* and *A. cruentus* during R, while *AhTPS9* was induced by both SWDS and R in all species tested with the exception of *A. hybridus*. *AhTPS11* expression was more sporadic and at lower intensity than in leaves. However, *AhTPS9 and AhTPS11* expression levels were higher in *A. hypochondriacus* and *A. cruentus* under SWDS (Figure 12B). Root *AhTPP* and *AhTRE* expression in response to WDS and R conditions remained mostly neutral or negative (Figure 12B). 

### 2.5. WDS Tolerance and Changes in SnRK1 and TOR Activity in Leaves and Roots

The rationale behind SnRK1 and TOR activity assays was to determine the effect of reduced energy supplies, typically caused by stress conditions, on the activity of these two master regulators of metabolism. While SnRK1 activates catabolism and represses energy-consuming anabolic processes in response to declining energy availability triggered by stress, TOR is known to promote growth via the integration of energy surplus with increased cell proliferation and development, protein synthesis, transcription, and metabolism [33,34]. In the amaranth species analyzed, SnRK1 activity was higher in roots. In *A. hypochondriacus*, the already high basal leaf and root SnRK1 activities were further increased by SWDS and MWDS, respectively (Figure 13A,B). In *A. caudatus* plants, basal SnRK1 activity was similar to *A. hypochondriacus*. However, WDS conditions did not alter their leaf SnRK1 activity, while SWDS conditions significantly reduced it in roots (Figure 13A,B). SWDS had a contrasting effect on SnRK1 activity in leaves and roots *A. hybridus*, while no changes were detected in *A. cruentus* (Figure 13A,B).

Basal TOR activities were higher in roots of WDS-tolerant amaranths. However, MWDS and SWDS led to ca. 3-fold reduction in TOR activity in roots of *A. hypochondriacus* (Figure 14A,B). Significantly reduced TOR activity was also observed in *A. caudatus* roots subjected to SWDS, while a ca. 3-fold increase in root TOR activity levels was detected under R (Figure 14B). No obvious correlation between SnRK1 activity and the expression levels of the *AhSnRK1α* and *AhGRIK2-like* genes was observed (Appendix A). These genes code for a kinase involved in the upstream activation of SnRK1 in plants and for the catalytic subunit of SnRK1, respectively [35,36]. 

A subsequent analysis was performed to test the effect of WDS on *AhSnRK2* orthologs of Arabidopsis *SnRK2* genes that function as positive regulators of ABA signaling [37]. Unexpectedly, the two *AhSnRK2* genes examined were repressed in leaves of the WDS-tolerant amaranth species, while WDS induced *AhSnRK2.2* in the leaves of *A. hybridus* plants (Appendix A).

### 2.6. WDS Tolerance and Changes in the Expression of ABA Signaling-Related Genes in Leaves and Roots

Owing to the critical function of ABA as a mediator of plant responses to drought [38], an analysis of genes associated with ABA signaling and response was performed. The expression of these ABA marker genes varied depending on the treatment applied, organ examined and species (Figure 15A,B). In leaves, the *AhDREB2A* transcription factor-encoding gene was induced by WDS and R in leaves of *A. hypochondriacus* plants only. On the other hand, a strong induction of the stress-adaptation *AhLEA14* gene [39] was produced in all genotypes and conditions tested, together with the *AhABI5* gene. The latter codes for a bZIP transcription factor that functions both as a regulator of abiotic stress responses and as an integrator of ABA crosstalk with other phytohormones [40], which was induced mostly under WDS. The latter results were indicative of the activation of ABA signaling in response to WDS in leaves of amaranth plants, although the expression of the *responsive to ABA18* gene (*AhRAB18*), usually detected in plants exposed to water stress or exogenous ABA [41] was very sporadically induced by WDS in leaves, and only in WDS-sensitive amaranth species (Figure 15A). All the above ABA-marker genes, including *AhRAB18*, were more strongly induced in roots of *A. hypochondriacus* (Figure 15B). 

## 3. Discussion

This study showed that *A. hypochondriacus*, a Mesoamerican grain amaranth, had superior WDS tolerance when compared to *A. cruentus*, another Mesoamerican grain amaranth, to *A. hybridus*, its proposed wild relative and to *A. caudatus*, the South American grain amaranth [10,11]. The salient finding of this study indicated that the enhanced WDS tolerance observed in *A. hypochondriacus* could be attributed to three key responses to WDS, including: (i) A better ability to establish an OA and advantageous sink-source relationships upon fast rates of dehydration; (ii) a stronger/more efficient SSR; and (iii) a heightened ABA response.

### 3.1. OA as a Mechanism Associated with Increased WDS in A. Hypochondriacus

OA is designed to maintain water balance in stressed plants by accumulating different osmolytes, including soluble NSCs, Pro, RFOs and others. These protect cellular structures and delay dehydration-related damage by maintaining cell turgor and other physiological mechanisms under water-deficit conditions [42]. Osmolytes accumulation in leaves, and more prominently in roots, of WDS-tolerant amaranth species, may have also counterbalanced a decrease in CO_2_ supply resulting from stomatal closure occurring in response to water loss [43]. Huerta-Ocampo et al. [15] provided the first indication that the accumulation of reducing sugars and Pro in roots was a contributing factor to WDS tolerance in *A. hypochondriacus*. Differential Pro and RFOs accumulation in response to WDS was also reported in tolerant genotypes of *Chenopodium quinoa*, a close relative of amaranth [44]. It remains to be determined if the OA established in roots of *A. hypochondriacus* and *A. cruentus* also enhanced root growth and soil moisture extraction as reported in other WDS-tolerant crops [26,42,45]. 

Data generated strongly suggested that the purposely more effective OA observed in the WDS tolerant amaranth genotypes was facilitated by a larger release of Hex from their starch reserves, coupled to a more intense accumulation of Raf and Pro in leaves and/or roots. A study comparing two maize genotypes having contrasting WDS-tolerance indicated that the tolerant genotype had ca. 3-fold higher root turgor under water limitation conditions [45]. This was attributed to a higher rate of sugar concentration in root apices, which inversely correlated with lower starch contents. WDS also led to a ca. 3-fold higher Pro concentration after a 72 h exposure in the WDS-tolerant maize genotype. A related study also found that drought-induced osmolyte accumulation was a factor that contributed to withstand WDS in tolerant maize hybrids [42]. 

The link between OA and favorable sink-source relationships in WDS-tolerant amaranth genotypes was probably established by the strong WDS-related depletion of the starch reserves in leaf and roots, a phenomenon which was particularly strong under SWDS conditions. The latter proposal is supported by the accelerated starch catabolism usually observed in response to various abiotic stresses, which leads to soluble NSCs accumulation. In tolerant grain amaranths, root starch depletion coincided with a WDS-induction of amylase activity, which probably played a role in the generation of the high Hex content and Hex/Suc ratio observed in this organ. The similar leaf amylase activity levels detected in WDS-tolerant and WDS-sensitive amaranths suggest that additional starch degradation mechanisms contributed to the marked starch depletion detected in leaves of WDS-tolerant amaranth species. [46].

The higher Hex content and Hex/Suc ratio observed in *A. hypochondriacus* roots also involved, apparently, augmented root CWI, VI, and SuSy activities induced in response to WDS. This modification shared similarities with the C mobilization pattern associated with WDS tolerance in rice [47]. Previous reports in grain amaranth also showed that the SSR triggered by severe defoliation led to an active mobilization of C reserves, mostly starch, which was facilitated by the induced activity of various sucroytic and amylolytic enzymes [48,49,50].

RFOs, such as Raf and Gol, are extensively distributed in higher plants. They are involved in dessication tolerance and can also function as plant C stores [25,51,52,53]. Biosynthesis of RFOs originates from Gol generated from MI and UDP-galactose by Gol synthase (GolS). Gol subsequently acts as a galactose unit donor to Suc to generate Raf, Sta, Ver and higher order RFOs, via their respective glycosyltransferases/synthases [54]. This study indicated that RFOs might have also contributed to the improved OA observed in tolerant amaranth genotypes. Relevant changes were the high leaf, and root contents of Raf detected under MWDS, and the foliar accumulation MI, Gol and Raf observed under SWDS in tolerant *A. hypochondriacus* and *A. cruentus*, respectively. Modified MI and/or Raf levels were also observed in leaves of two contrasting quinoa genotypes subjected to WDS [44], and in a WDS-tolerant alfalfa cultivar able to accumulate Raf and Gol in roots during stress [55]. Significantly higher *AhGolS1* and *AhRafS* expression levels in WDS-tolerant grain amaranths, especially under SWDS, were somehow consistent with their higher RFOs content. These results were in accordance with a study of *Coffea canephora* clones with contrasting tolerance to WDS, in which the expression of *CcGolS1* was strongly repressed in WDS-sensitive clones [56], and with a companion report in *C. arabica* showing that the *CaGolS1* isoform was highly responsive to WDS [57]. Several other studies reported that the expression of *GolS* genes was congruous with abiotic stress tolerance in Arabidopsis [51,58] and in transgenic tobacco plants [59,60].

Although the accumulation of Sta and Ver in roots of WDS-stressed amaranth plants was in accordance with their reported increase in stressed leaves of *C. arabica* [57], it did not seem to contribute to WDS tolerance in grain amaranth. However, other results, suggest that presumptive RFOs with a higher degree of polymerization differentially accumulated in leaves WDS-tolerant amaranths. It is tempting to speculate that their accumulation was also a contributing factor to their observed tolerance to WDS. Although the possibility remains to be determined, it is supported by studies whose data suggested that Gol accumulation in water-stressed coffee was funneled to the generation of larger stress-protective RFOs by unidentified glycosyltransferases [56,57]. It might also provide an explanation why the intense induction of some RFO biosynthetic genes in roots of stressed amaranth plants, particularly in WDS-tolerant species did not translate into correspondingly higher Raf contents and, perhaps, other RFOs. These could have been used, instead, as precursors for the biosynthesis of these presumably more complex RFOs.

Pro is proposed to participate in the stabilization of sub-cellular structures, free radicals scavenging, and the regulation of cellular redox potential, among others. Together with other osmolytes, Pro is also known to promote plant recovery from drought stress by enhancing the capacity to avoid or repair membrane damage and to maintain membrane stability during dehydration and rehydration processes [25,61,62]. Thus, significantly higher Pro content most probably contributed to the improved OA capacity in leaves of *A. hypochondriacus* and *A. cruentus*, principally during SWDS. The idea is supported by previous studies showing that a differential Pro accumulation was observed in leaves of WDS tolerant genotypes of quinoa [44,63] and alfalfa [55]. The subsequent decline in Pro levels observed in most species during R, except in roots of *A. hybridus*, coincided with findings reporting that Pro is rapidly metabolized in order to provide N and reducing power during stress recovery processes [62,64]. 

Although present in trace amounts in most plants, Tre is another osmolyte that enables proteins to maintain their hydration state and thereby contribute to abiotic stress resistance [25,27]. In this study, however, WDS-responsive Tre accumulation in roots and leaves was similar in WDS-tolerant and WDS-sensitive species. This outcome was partly in agreement with data showing that Tre did not protect yeast cells from desiccation [65] and with reports that found no link between increased Tre accumulation and stress tolerance in plants or that showed that increased Tre levels in certain Arabidopsis mutants increased their WDS sensitivity [66]. 

### 3.2. A More Robust SSR May Contribute to Increased WDS in A. hypochondriacus

A second factor advocated to contribute to WDS tolerance in *A. hypochondriacus* was a more efficient regulation of the SSR via SnRK1 and possibly other metabolic controllers. The latter considering that only in this amaranth species did the activity of the SnRK1 and TOR plant metabolic regulators become altered under energy-deficient conditions. This was in accordance with the “low energy” syndrome caused by WDS in both leaves and roots. Variations of SnRK1 activity in *A. hypochondriacus* correlated to a certain degree with the expression patterns of the *AhGRIK2-like* and *AhSnRK1α* genes in roots, only. This scenario is further favored by the unresponsiveness to WDS shown by the *AhTPS1* gene, required for the synthesis of T6P—a plant signaling sugar that inhibits SnRK1 [27,29,67,68]. Likewise, a connection between certain class II *TPS* genes and WDS tolerance in *A. hypochondriacus* was suggested by: (i) A general downregulation of foliar *AhTPS5* during WDS and R and of several other *class II TPS* genes during R; (ii) the high expression of *AhTPS9* and *AhTPS11* observed in leaves and roots during SWDS; (iii) the down-regulation of the *AhTPPD* and *AhTPPI* genes during WDS; and (iv) the induction of foliar *AhTRE* during WDS. Past studies have shown that class II TPS proteins have differential sensitivity to Suc levels in plants [69], which was important in the context of modified NSCs content observed in response to WDS in amaranth. The above results were also consistent with the upregulation of *TPS11* and *TRE* detected in stomatal guard cells of sucrose-treated Arabidopsis [70], considered to be part of a connection between Tre, class II TPSs, carbohydrate metabolism regulation, and stomatal movements via sugar sensing. The significantly higher upregulation of *AhTPS9* and *AhTPS11* under SWDS in sucrose-depleted roots of *A. hypochondriacus* plants was also in accordance with studies showing that Suc-limiting conditions, led to the induction of the *AtTPS8-AtTPS11* genes in Arabidopsis [33,71], and with the finding that *TPS6, TPS8, TPS9, TPS10 and TPS11* were part of the genes recognized as belonging to the SSR network [72]. Also relevant is the finding that the overexpression of *OsTPS9* in rice significantly increased tolerance toward cold and salinity stress through its proposed association with OsTPS1 [73]. On the other hand, the general WDS-unresponsiveness of *class II TPS*, *TPP* and *TRE* genes in *A. caudatus* and *A. hybridus* could have contributed to their WDS sensitivity.

### 3.3. Stronger ABA-Related Gene Expression Coincides with Increased WDS Tolerance in A. hypochondriacus

Increased WDS tolerance in grain amaranth also appeared to involve a heightened ABA-related gene expression in response to water loss. It may be hypothesized that the positive modulation of ABA-related genes contributed to enhanced OA, via the increased Glu levels detected in tolerant grain amaranths, similarly to what was previously reported in Arabidopsis seedlings [74]. Differences in ABA content and/or sensitivity were also found to be associated with WDS tolerance in alfalfa [55]. 

Two of the above genes are considered to have a prominent role in ABA signaling: (i) *AhABI5*, which acts as an integrator of the frequently antagonistic WDS-activated phytohormone signaling pathways, to ensure an adequate response to different WDS intensities [40]; and (ii) *AhDREB2A*, coding for AhDREB2A, a member of the group IV/A-2 subfamily of AP2/ERF proteins considered to form a central response network to control diverse abiotic stresses. Thus, AhDREB2A is proposed to regulate ABA crosstalk with other signaling pathways. AhDREB2A is also presumed to play a pivotal role in ABA-independent responses [2,75]. Moreover, previous data obtained from *abi5* legume mutants [76], suggest the possibility that induced *AhABI5* expression may have contributed to the augmented expression of *AhRafS* detected in leaves and roots of WDS tolerant amaranths. *LEA* genes code for a family of protective late embryogenesis abundant proteins that shield macromolecules and/or various cellular structures during WDS by generating a hydration shell that maintains the integrity and function of their targets [77]. Recently, a role for stress amelioration was attributed to an *A. cruentus* LEA protein which was found to accumulate in several organs of amaranth plants in response to salt stress [78], while a sweet-potato LEA14 positively influenced lignification and increased osmotic-, heat- and/or salt stress-tolerance in transgenic sweet-potato calli [39] and poplar plants [79]. However, the general WDS unresponsiveness of the *SnRK2* subgroup genes analyzed in this study was intriguing, considering their proposed role in stress amelioration, partly through their involvement in the ABA signaling pathway [67,80].

### 3.4. Final Observations

However, consideration must be made that the data generated in this study, although revealing of several biochemical and molecular traits associated with WDT, should be complemented with additional experimentation to address a number of limitations inherent to the experimental design utilized [81]. The use of short duration experiments did not allow to determine WDS tolerance at other development stages, particularly in the reproductive stages, critical for yield. This limitation also excluded the possible contribution of developmentally-related changes to WDS tolerance, such as leaf expansion, associated with leaf area (LA) and leaf longevity/senescence. The former is important, considering that LA is a crucial determinant of total water loss from a plant. Plant architecture is another important aspect associated with development that was overlooked in this study. This factor is known to influence plant water relations by determining leaf exposure to light and air movement and, consequently, the transpiration rate/ unit LA. Additionally, the results of this study were generated by soil water depletion experiments in which the plants were grown individually in pots scaled to the plant size and filled with a rich organic substrate. Here, the water content of the soil in a pot was the starting experimental baseline. Limitations associated with this method are the frequently poor relationship established between the soil water content and the plant water status. Another disadvantage is that the root zone usually dries unevenly from the soil surface down. Soil density, not generally controlled, may also contribute to variation in plant responses. Moreover, the use of limited volumes of soil leads to a rapid depletion of water, therefore greatly favoring regulation of plant water balance by decreased stomatal conductance and minimizing the contribution of reduced LA or loss of functional leaves. The latter has greater relevance in more biologically relevant field conditions—where water loss is slower, and there is a larger volume of soil to be exploited by an expanding root system. Lastly, this study needs to be complemented with others able to assess WDS tolerance in grain amaranth under a wider range of water availability and ambient conditions and to determine the ultimate correlation between WDS tolerance and total biomass and/or yield.

## 4. Materials and Methods

### 4.1. Plant Material

Three grain amaranth species (*A. hypochondriacus*, *A. cruentus* and *A. caudatus*) together with vegetable amaranth (*A. hybridus*), believed to be grain amaranths’ ancestor [10,11], were employed in the greenhouse experiments here described. All plant materials were provided by Dr. Eduardo Espitia Rangel, INIFAP, México, curator of the Mexican amaranth germplasm collection. Approximately three week-old plants, which all had 10–12 expanded leaves, were employed for experimentation. The plants were germinated in a conditioned growth chamber, as described [82]. The seedlings were then transferred 1.3 L plastic pots containing 250 g of a general substrate and grown in a greenhouse under natural conditions of light and temperature (see below). A total of 8 cultivars/accessions of at least one of the above species was tested, as follows: *A. hypochondriacus* (“Gabriela”, “Revancha” and “DGTA” cultivars); *A. cruentus* (“Amaranteca”, “Dorada” and “Tarasca” cultivars), *A. caudatus* (no classification available) and *A. hybridus* (accession N°. 1330). 

### 4.2. Water-Deficit Stress (WDS) Experiments

All WDS experiments were performed in a commercial greenhouse with zenithal and lateral type ventilation (Baticenital 850; ACEA S.A., Santa Irene, Texcoco, Mexico) from May to August, 2015. The average temperatures in the greenhouse ranged between 15 °C (night) and 38 °C (day), with an average 55% R.H. The experiments were performed under natural light and photoperiod (photosynthetic photon flow density ≈ 1300 µE, ≥ 12 h light). An initial experiment was performed to screen the eight cultivars/accessions mentioned above for their tolerance to WDS, using a randomized block design. WDS was established by withholding watering for 5 or 10 days, time after which moderate to severe plant wilting was evident. Ten plants per species per cultivar/accession were used for each treatment. WDS tolerance was scored using two parameters: (i) Leaf water potentials (ψ_L_, in MPa), determined in leaf discs at the end of the two WDS treatments described, using a WP4 Dewpoint PotentiaMeter (Decagon Devices Inc., Pullman, WA, USA) as instructed by the manufacturers; and (ii) percentage of recovery, determined in whole plants one day after normal watering was restored following each WDS stress. This information led to the selection of 4 materials for subsequent experimentation. These were the following: *A. hypochondriacus* (var. “Gabriela”) and *A. cruentus* (var. “Amaranteca”), classified as “WDS-tolerant”, and *A. caudatus* and *A. hybridus*, as “WDS-susceptible”. Subsequently, two tandem experiments were performed in the above conditions to test WDS tolerance based on soil water content depletion. Prior to the start of the WDS trials, each experimental 1.3 L pot was weighed individually at soil field capacity (FC). WDS trials were started when all pots were at 90% FC. Control plants were kept in these conditions for the duration of the experiments, whereas WDS was established by withholding watering until the SWC in each pot reached either 30% FC (“moderate WDS, [MWDS]”) or 10% FC (“severe WDS, [SWDS]”). These stress levels were reached approximately 5–6 and 9–10 days after regular watering was withheld, respectively. An additional group of plants was re-watered after reaching 10% FC and was allowed to recover for 24 h (“recovery”, [R]). The pots were weighed daily to determine water loss—taking care to ensure it was consistently done at approximately the same time of the day. Twelve plants were used per treatment using a randomized block design, as above. The roots and all fully expanded leaves from 4 similarly treated plants were sampled. Roots and leaves of four plants were combined into a single pooled sample. Control plants were similarly sampled. This procedure generated three root and leaf biological replicates per species, treatment and experiment, each comprised of pooled tissues from four plants. The experiment was replicated twice during the period mentioned, with similar results. All pooled tissue samples were flash frozen in liquid N_2_ and stored at −70 °C until needed. At least three technical replicates of each biological sample were included in the biochemical and molecular assays described below. 

### 4.3. Extraction of Total RNA and Gene Expression Analysis by RT-qPCR

Total RNA was extracted from frozen leaf and root tissue using the Trizol reagent (Invitrogen, Carlsbad, CA, USA) as instructed, except for the addition of a salt solution (sodium citrate 0.8 M + 1.2 M NaCl) during precipitation in a 1: 1 *v*/*v* ratio with isopropanol and a further purification step with LiCl (8 M) for 1 h at 4 °C [83]. Quantitative gene expression analysis using *SYBR Green* detection chemistry (Bio-Rad, Hercules, CA, USA) was performed as described previously [83]. Primers designed for the amplification of the pertinent amaranth gene transcripts was performed according to Thornton and Basu [84] and was based on recently published genomic data [31] (Appendix A). Relative gene expression was calculated using the comparative cycle threshold method [28] using the *AhACT7*, *AhEF1a* and *AhβTub5* genes for data normalization.

### 4.4. Determination of NSC and Pro

Soluble NSCs and Pro contents were quantified in leaf and root samples, according to Palmeros-Suárez et al. [83]. Briefly, soluble NSC (glucose + fructose + sucrose) content was determined using an enzymatic coupling method designed to specifically quantify each NSC on the basis of a 1: 1 hexose: NADPH stoichiometry. Insoluble starch was hydrolyzed to free glucose by a mixture of α-amylases and amyloglucosidases. It was subsequently quantified on the basis of the free glucose released, as described above. Proline was assayed colorimetrically, at 520 nm, by measuring the red chromogen produced by its reaction with ninhydrin at low pH. 

### 4.5. Determination of RFOs by HPAEC–PAD

Identification and determination of RFOs content in leaf and root samples were performed by High-Performance Anion-Exchange Chromatography with Pulsed Amperometric Detection (HPAEC–PAD), according to Mellado-Mojica et al. [85]. All chemicals used for the optimization of the chromatographic separation conditions and for quantitation were acquired from Sigma (Sigma-Aldrich, St. Louis, MO, USA). These were the following: MI, Gol, Raf, Sta and Ver.

### 4.6. Determination of Tre 

Tre was determined by GC/MS using an ion selective method, as described previously [86]. Briefly, sugars in water soluble leaf and root extracts were enriched by their passage through a cationic exchange chromatography column able to retain charged compounds, such as amino acids and organic acids. The flow-through sugar solution was vacuum-evaporated to dryness, re-dissolved in a reaction mixture containing 20 µL pyridine and 80 µL of the derivatization N, O-bis (trimethylsilyl)-trifluoroacetamide reagent and incubated at 80 °C for 30 min. One µL aliquots of the reaction solutions were analyzed by a gas chromatography system coupled to a mass-selective detector using the conditions and parameters specified by the authors. 

### 4.7. Determination of Invertases, Sucrose Synthase and Amylase Activities

Vacuolar (VI), cell wall (CWI), and cytoplasmic (CI) invertases and sucrose synthase (SuSy) activities were determined as described by Wright et al. [87]. Briefly, frozen leaf and root samples were extracted in specified extraction buffers and centrifuged. The soluble portion was used to measure vacuolar and cytoplasmic invertase activities, while the pellet was employed for cell wall invertase determinations under the conditions of pH and temperature described by the authors. The respective invertase activities were calculated by measuring the amount of glucose and fructose released, which were quantified using the enzymatic-coupling assay described above. Sucrose synthase activity was similarly determined, in the sucrose hydrolysis direction, by measuring the amount of UDP glucose produced, which was equivalent to the NADPH released as a result of its oxidation by UDP-glucose dehydrogenase. Amylase activity was determined according to Bernfeld [88]. This method measured the rate at which maltose was released from starch by monitoring the concomitant reduction of 3, 5-dinitrosalicylic acid, at 540 nm. One unit was defined as the amylase activity needed to release one micromole of β-maltose per min at 25 °C and pH 4.8 under the specified conditions. All assays were modified to fit a micro-plate format.

### 4.8. Determination of SnRK1 and TOR Protein Kinase Activities

The SnRK1 activity was determined using the specific AMARA peptide substrate previously reported by Dale et al. [89]. Briefly, 1 µL aliquots of plant tissue extracts were placed in fresh Eppendorf tubes and mixed with 19 µL of 40 mM HEPES buffer solution, pH 7.5, containing 5 mM MgCl_2_, 200 uM de [γ-^32^P] ATP (3.77 TBq mol^−1^), 200 µM of the AMARA peptide and 5 mM DTT in addition to 1 µM okadaic acid (a phosphatase inhibitor) and 1 µM pepstatin A, 10 µM E64 and 7 µM chimostatin (proteinase inhibitors). The reaction mixture was incubated for 10 min at room temperature and then stopped by adding a phospho-cellulose paper strip able to bind the positively charged peptide. The paper strip was washed thrice with 75 mM phosphoric acid by shaking in a tube mixer for 15, 10, and 10 min intervals, respectively. The washing solutions were removed accordingly using a vacuum pump. The paper strips were given a 5 min final wash with 1 mL molecular grade absolute ethanol and then dried at room temperature for 10–15 min. Four paper strips not subjected to the washing procedure and previously exposed, two each, to 2.5 µL (0.5 nmol of [γ-^32^P] ATP) or 5.0 µL (1 nmol of [γ-^32^P] ATP) were used as measures of the total radioactivity present in the reaction mixtures. Radioactivity in the control and reaction paper strips was determined in a 5000CE liquid scintillation counter. (Beckman, Brea, CA, USA). Kinase activity was calculated using the resulting radioactivity readings and was reported in a total protein basis. The latter was determined with using the Bradford method [90] employing a commercial kit (Bio-Rad).

This methodology was adapted to measure TOR kinase activity. The only difference was the use of a different peptide substrate for TOR kinase, designed *de novo*. The consensus sequence of the phosphorylation site of the Ribosomal protein S6 Kinase (S6K) recognized by TOR kinase was used for this purpose. Peptide design involved alignment of diverse eukaryotic S6K proteins was performed using the *Multiple Sequence Alignment by Log-Expectation*, or MUSCLE, program of the MEGA7 platform [91]. The aligned sequences chosen for peptide design were those conserved in all eukaryotic organisms analyzed, including *Homo sapiens, Saccharomyces cerevisiae, Ratus norvegicus* and *Caenorhabditis elegans*, *A. thaliana, A. hypochondriacus* and all additional plant genome sequences reported in the Phytozome platform (https://phytozome.jgi.doe.gov/). Final modifications included the substitution of certain amino acids for alanine residues, and the addition of three arginine tags at its C-terminal end, to ensure binding to the phospho-cellulose paper strips. The final version of the peptide used for TOR kinase assays was AFAGFTYVAPRRR. Both this and the AMARA peptides were synthesized by Peptide 2.0 (https://www.peptide2.com/).

### 4.9. Statistical Analysis

All experiments were conducted using a randomized complete block design. One-way ANOVAs were utilized to evaluate the differences between treatment means. For ANOVAs where the F test was significant at *p* ≤ 0.05, the Tukey-Kramer test was applied. Statistical analysis was performed with R software (Development Core Team, https://www.r-project.org/). The R pheatmap package was used for the heat-map analysis. For box-plot analysis, the ggplot2, agricolae and reshape2 R packages were used.

## 5. Conclusions 

This study revealed that differential WDS tolerance between grain amaranths and *A. hybridus*, was likely due to quantitative differences in three key aspects associated with WDS amelioration responses: OA, ABA-related gene expression, and SSR. It is proposed that different responsiveness of the regulatory networks responsible of modulating WDS responses in grain amaranths and *A. hybridus* caused the variance in WDS tolerance observed between these amaranth species. Published transcriptomic data [82] revealed that several genes belonging to various categories associated with stress responses showed modified expression levels in *A. hypochondriacus* during WDS (Appendix A). Some of them supported the findings of this study, including the enrichment of sugar transporter genes and others, coding for carbohydrate metabolism enzymes, such as β-amylases and other starch degrading enzymes. The up-regulated expression of *GLUTAMATE SYNTHASE*, leading to the glutamate precursor of one of the proline biosynthetic pathways, was in agreement with WDS-induced proline accumulation. These may have reinforced the OA response believed to contribute to increased WDS tolerance in *A. hypochondriacus*. In addition, transcript accumulation of *NITRITE REDUCTASE,* several *MALATE DEHYDROGENASE* genes and a *UREASE ACCESSORY PROTEIN G*, coding for proteins involved in the coordination of nitrogen and carbon metabolism, partitioning of carbon and energy in leaves and nitrogen recycling, respectively, coincided with a more efficient SSR. Other categories suggest an active phytohormone cross-talk that could have modulated ABA responses. Transcritomic data also suggested that in addition to stronger OA, SSR and ABA-related responses, WDS tolerance in *A. hypochondriacus* might have resulted from the additional contribution of activated key metabolic processes, such as glycolysis, photosynthesis, antioxidant activity, cuticular wax synthesis and membrane lipid desaturation. Conversely, WDS led to a general down-regulation of a variety of transcription factor genes and several others involved in cell wall generation and maintenance. These genes are, therefore, candidates to conform the hypothetical network, responsible for the increased WDS tolerance observed in *A. hypochondriacus*. This is a possibility that remains to be validated by further experimentation. 

## Figures and Tables

**Figure 1 plants-08-00239-f001:**
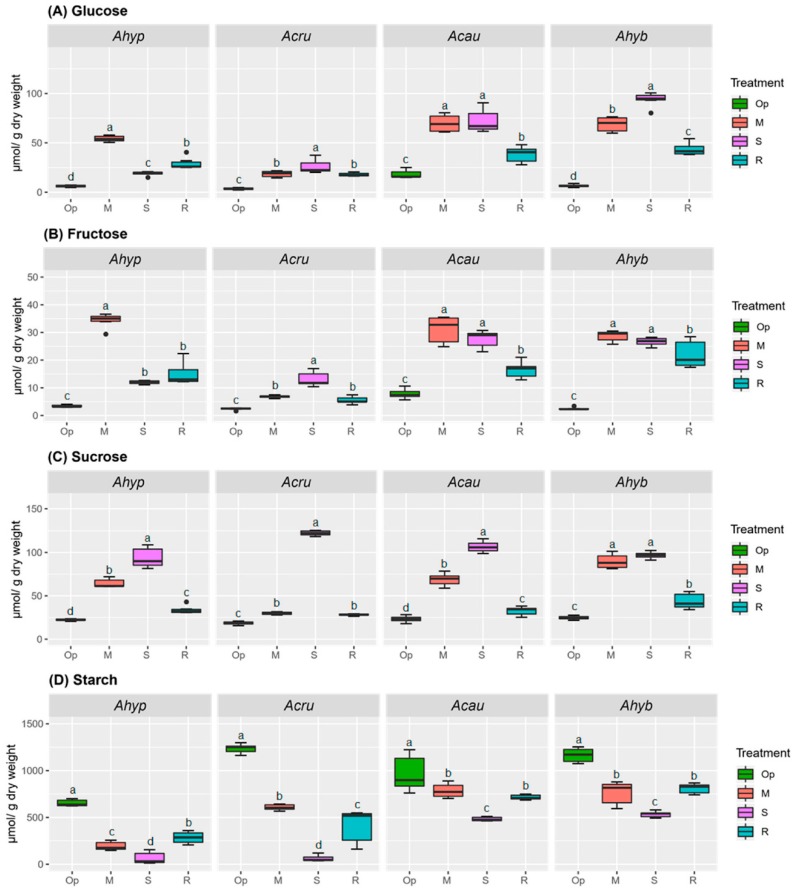
Non-structural carbohydrates in leaves of amaranth plants subjected to water-deficit stress. (**A**) Glucose, (**B**) fructose, (**C**) sucrose and (**D**) starch content in leaves of *Amaranthus hypochondriacus* (Ahypo), *A. cruentus* (Acru), *A. caudatus* (Acau) and *A. hybridus* (Ahyb) plants growing in optimal conditions (Op), subjected to moderate (M) or severe (S) water-deficit stress, or allowed to recover from S, 1 day after normal watering was restored (R). Different letters over the box-and-whisker plots represent statistically significant differences at *p* ≤ 0.05 (Tukey Kramer test; *n* = 9). Box-and-whisker plots show high, low, and median values. The results shown are those obtained from a representative experiment that was repeated in the spring-summer and summer-autumn seasons of 2015, respectively, with similar results.

**Figure 2 plants-08-00239-f002:**
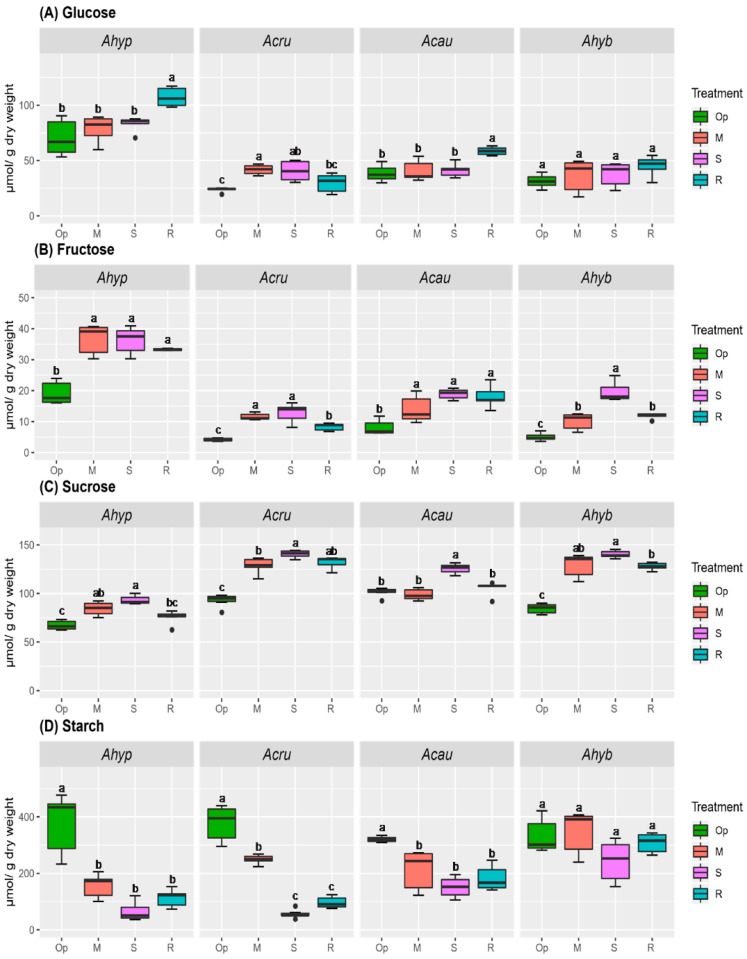
Non-structural carbohydrates in roots of amaranth plants subjected to water-deficit stress. (**A**) Glucose, (**B**) fructose, (**C**) sucrose and (**D**) starch content in roots of *Amaranthus hypochondriacus* (Ahypo), *A. cruentus* (Acru), *A. caudatus* (Acau) and *A. hybridus* (Ahyb) plants growing in optimal conditions (Op), subjected to moderate (M) or severe (S) water-deficit stress, or allowed to recover from S, 1 day after normal watering was restored (R). Different letters over the box-and-whisker plots represent statistically significant differences at *p* ≤ 0.05 (Tukey Kramer test; *n* = 9). Box-and-whisker plots show high, low, and median values. The results shown are those obtained from a representative experiment that was repeated in the spring-summer and summer-autumn seasons of 2015, respectively, with similar results.

**Figure 3 plants-08-00239-f003:**
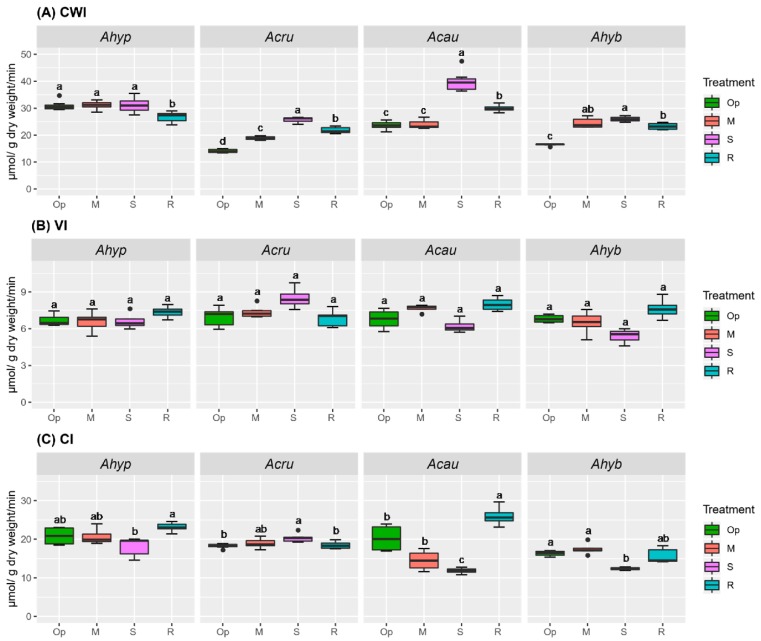
Invertase activity in leaves of amaranth plants subjected to water-deficit stress. (**A**) Cell wall invertase [CWI], (**B**) vacuolar invertase [VI], and (**C**) alkaline/neutral cytoplasmic invertase [CI] activities in leaves of *Amaranthus hypochondriacus* (Ahypo), *A. cruentus* (Acru), *A. caudatus* (Acau), and *A. hybridus* (Ahyb) plants growing in optimal conditions (Op), subjected to moderate (M) or severe (S) water-deficit stress, or allowed to recover from S, 1 day after normal watering was restored (R). Different letters over the box-and-whisker plots represent statistically significant differences at *p* ≤ 0.05 (Tukey Kramer test; *n* = 9). Box-and-whisker plots show high, low, and median values. The results shown are those obtained from a representative experiment that was repeated in the spring-summer and summer-autumn seasons of 2015, respectively, with similar results.

**Figure 4 plants-08-00239-f004:**
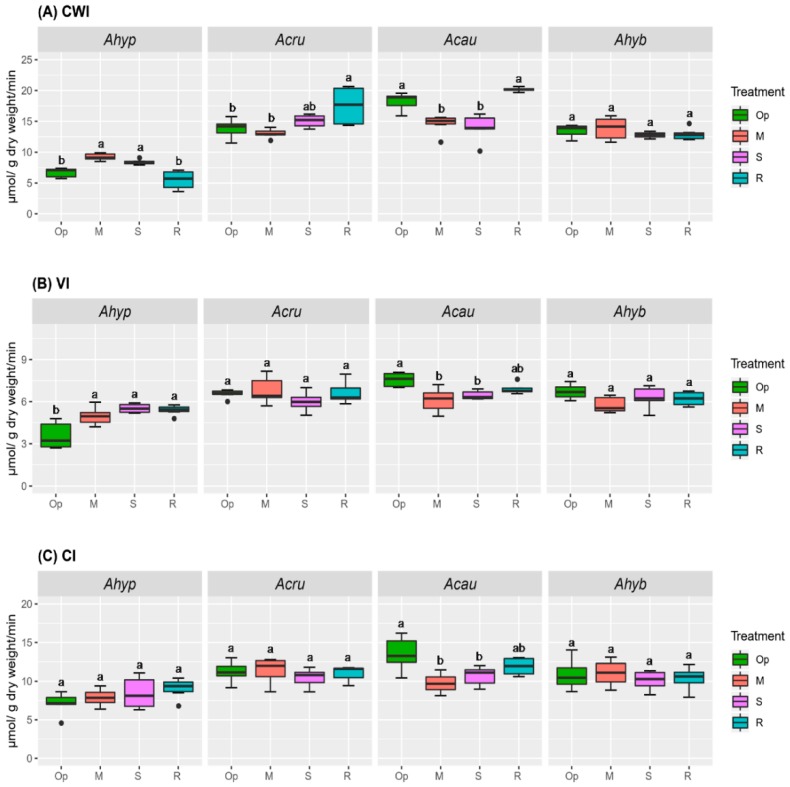
Invertase activity in roots of amaranth plants subjected to water-deficit stress. (**A**) Cell wall invertase [CWI], (**B**) vacuolar invertase [VI], and (**C**) alkaline/neutral cytoplasmic invertase [CI] activities in roots of *Amaranthus hypochondriacus* (Ahypo), *A. cruentus* (Acru), *A. caudatus* (Acau), and *A. hybridus* (Ahyb) plants growing in optimal conditions (Op), subjected to moderate (M) or severe (S) water-deficit stress, or allowed to recover from S, one day after normal watering was restored (R). Different letters over the box-and-whisker plots represent statistically significant differences at *p* ≤ 0.05 (Tukey Kramer test; *n* = 9). Box-and-whisker plots show high, low, and median values. The results shown are those obtained from a representative experiment that was repeated in the spring-summer and summer-autumn seasons of 2015, respectively, with similar results.

**Figure 5 plants-08-00239-f005:**
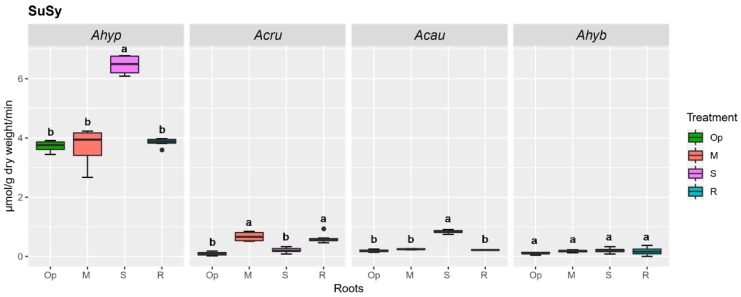
Sucrose synthase (SuSy) activity in roots of amaranth plants subjected to water-deficit stress. SuSy activity in roots of *Amaranthus hypochondriacus* (Ahypo), *A. cruentus* (Acru), *A. caudatus* (Acau), and *A. hybridus* (Ahyb) plants growing in optimal conditions (Op), subjected to moderate (M) or severe (S) water-deficit stress, or allowed to recover from S, 1 day after normal watering was restored (R). Different letters over the box-and-whisker plots represent statistically significant differences at *p* ≤ 0.05 (Tukey Kramer test; *n* = 9). Box-and-whisker plots show high, low, and median values. The results shown are those obtained from a representative experiment that was repeated in the spring-summer and summer-autumn seasons of 2015, respectively, with similar results.

**Figure 6 plants-08-00239-f006:**
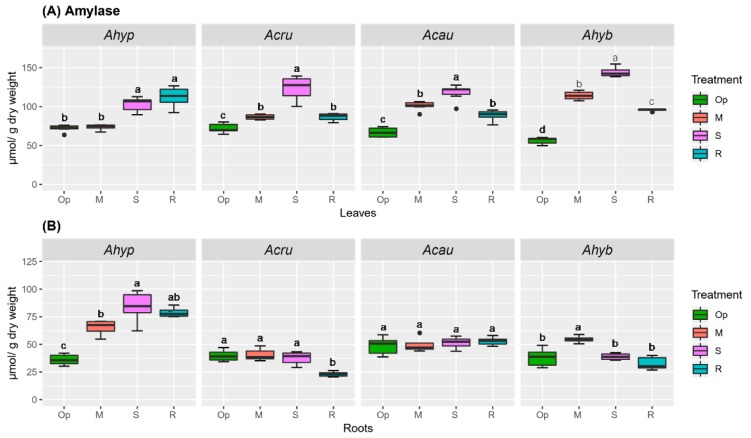
Amylase activity in amaranth plants subjected to water-deficit stress. Amylase activity in (**A**) leaves and (**B**) roots of *Amaranthus hypochondriacus* (Ahypo), *A. cruentus* (Acru), *A. caudatus* (Acau) and *A. hybridus* (Ahyb) plants growing in optimal conditions (Op), subjected to moderate (M) or severe (S) water-deficit stress, or allowed to recover from S, 1 day after normal watering was restored (R). Different letters over the box-and-whisker plots represent statistically significant differences at *p* ≤ 0.05 (Tukey Kramer test; *n* = 9). Box-and-whisker plots show high, low, and median values. The results shown are those obtained from a representative experiment that was repeated in the spring-summer and summer-autumn seasons of 2015, respectively, with similar results.

**Figure 7 plants-08-00239-f007:**
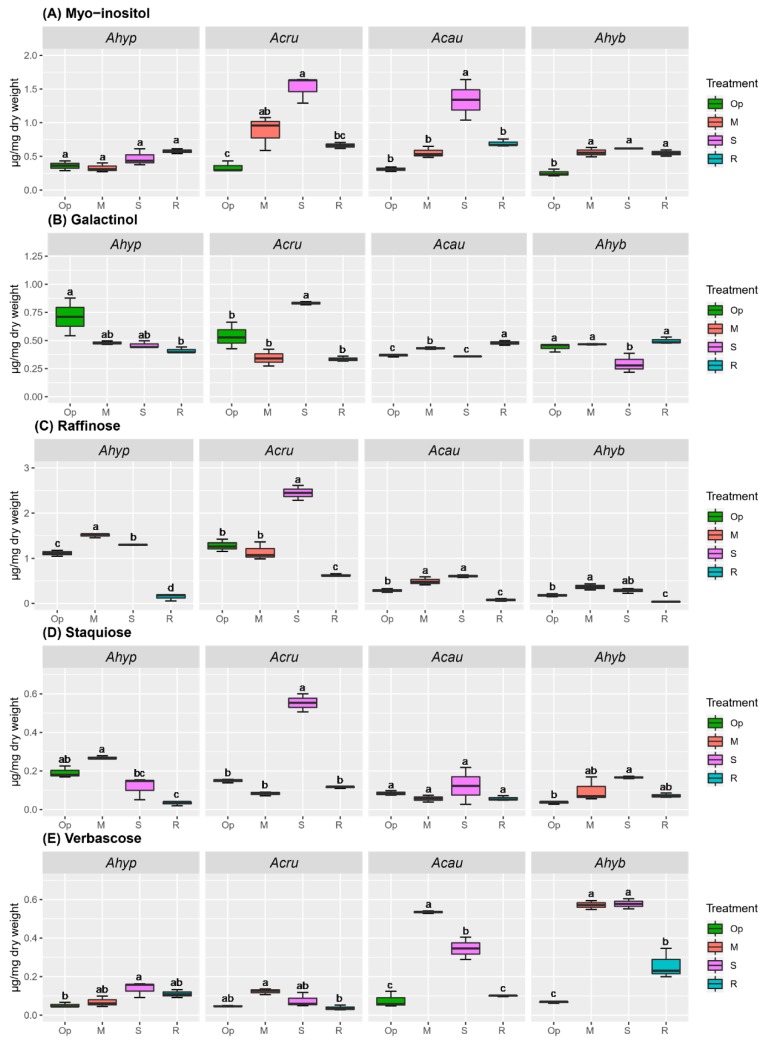
Raffinose family oligosaccharides (RFOs) in leaves of amaranth plants subjected to water-deficit stress. RFOs in leaves of *Amaranthus hypochondriacus* (Ahypo), *A. cruentus* (Acru), *A. caudatus* (Acau), and *A. hybridus* (Ahyb) plants growing in optimal conditions (Op), subjected to moderate (M) or severe (S) water deficit stress, or allowed to recover from S, 1 day after normal watering was restored (R). RFOs and RFO precursors analyzed were: (**A**) Myo-inositol; (**B**) galactinol; (**C**) raffinose; (**D**) staquiose; and (**E**) verbascose. Different letters over the box-and-whisker plots represent statistically significant differences at *p* ≤ 0.05 (Tukey Kramer test; *n* = 9). Box-and-whisker plots show high, low, and median values. The results shown are those obtained from a representative experiment that was repeated in the spring-summer and summer-autumn seasons of 2015, respectively, with similar results.

**Figure 8 plants-08-00239-f008:**
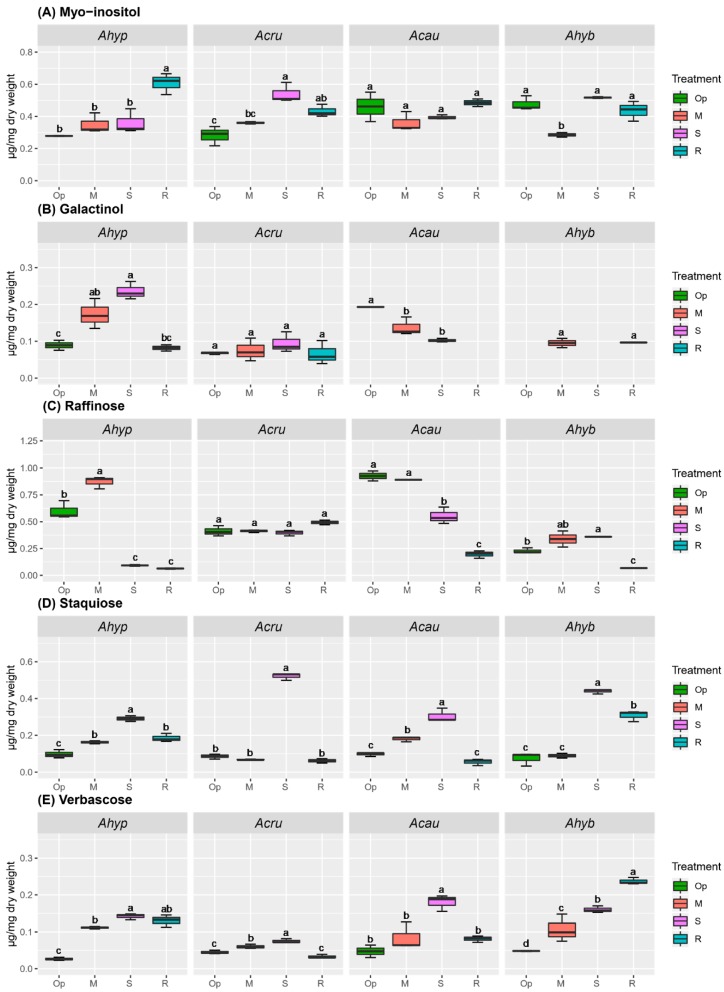
Raffinose family oligosaccharides (RFOs) in roots of amaranth plants subjected to water-deficit stress. RFOs in roots of *Amaranthus hypochondriacus* (Ahypo), *A. cruentus* (Acru), *A. caudatus* (Acau), and *A. hybridus* (Ahyb) plants growing in optimal conditions (Op), subjected to moderate (M) or severe (S) water deficit stress, or allowed to recover from S, 1 day after normal watering was restored (R). RFOs and RFO precursors analyzed were: (**A**) Myo-inositol; (**B**) galactinol; (**C**) raffinose; (**D**) staquiose; and (**E**) verbascose. Different letters over the box-and-whisker plots represent statistically significant differences at *p* ≤ 0.05 (Tukey Kramer test; *n* = 9). Box-and-whisker plots show high, low, and median values. The results shown are those obtained from a representative experiment that was repeated in the spring-summer and summer-autumn seasons of 2015, respectively, with similar results.

**Figure 9 plants-08-00239-f009:**
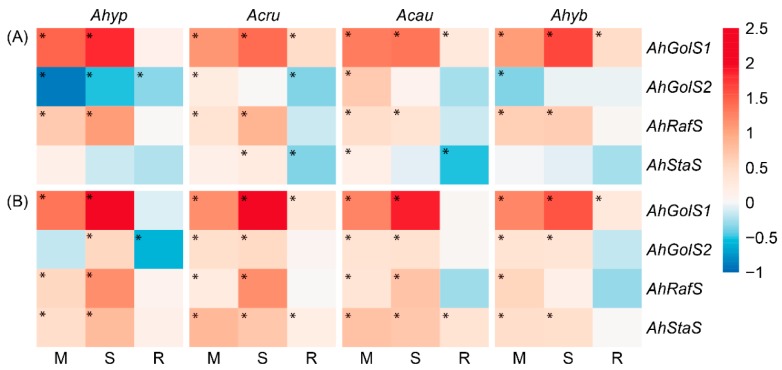
Relative expression of genes involved in the biosynthesis of raffinose family oligosaccharides in (**A**) leaves and (**B**) roots of *Amaranthus hypochondriacus* (Ahypo), *A. cruentus* (Acru), *A. caudatus* (Acau), and *A. hybridus* (Ahyb) plants subjected to moderate (M) or severe (S) water deficit stress, or allowed to recover from S, 1 day after normal watering was restored (R). Induced (normalized expression values ≥ 1.5; in red) and repressed (normalized expression values ≤ 0.5; in blue) gene expression values, represented in a log_10_ basis, are highlighted with an asterisk at the upper left side of the cells. They were calculated according to the comparative cycle threshold method [28] using the *AhACT7*, *AhEF1a* and *AhβTub5* amaranth genes for data normalization.

**Figure 10 plants-08-00239-f010:**
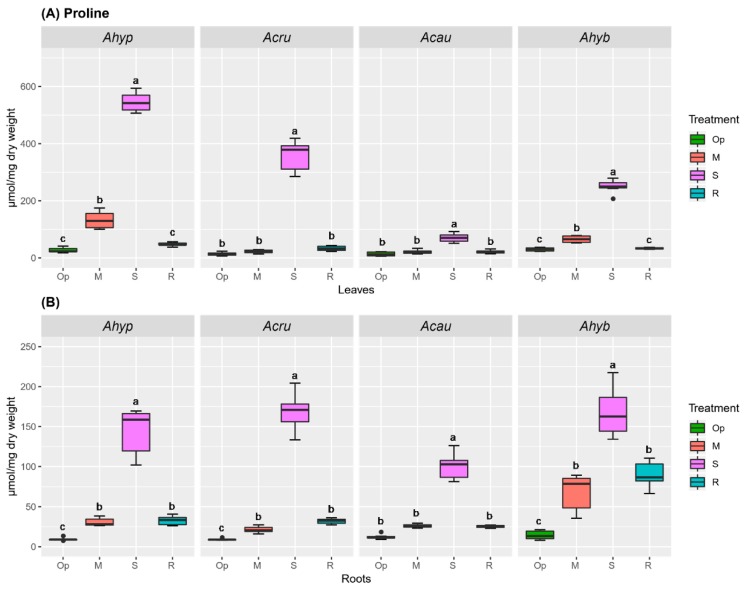
Proline (Pro) accumulation in amaranth plants subjected to water-deficit stress. Pro content in (**A**) leaves and (**B**) roots of *Amaranthus hypochondriacus* (Ahypo), *A. cruentus* (Acru), *A. caudatus* (Acau) and *A. hybridus* (Ahyb] plants growing in optimal conditions (Op), subjected to moderate (M) or severe (S) water deficit stress, or allowed to recover from S, 1 day after normal watering was restored (R). Different letters over the box-and-whisker plots represent statistically significant differences at *p* ≤ 0.05 (Tukey Kramer test; *n* = 9). Box-and-whisker plots show high, low, and median values. The results shown are those obtained from a representative experiment that was repeated in the spring-summer and summer-autumn seasons of 2015, respectively, with similar results.

**Figure 11 plants-08-00239-f011:**
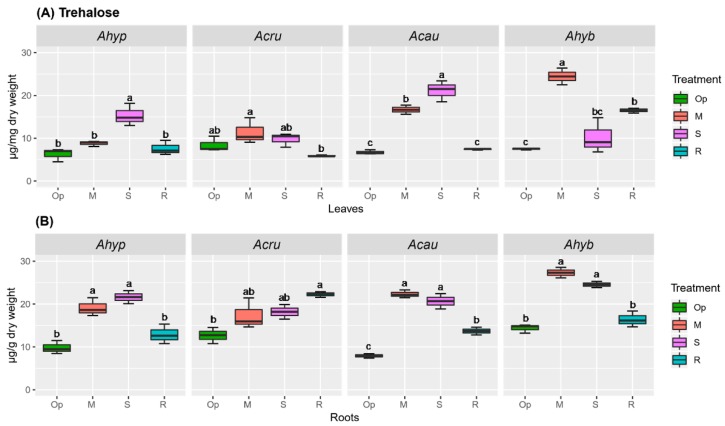
Trehalose (Tre) accumulation in amaranth plants subjected to water-deficit stress. Tre content in (**A**) leaves and (**B**) roots of *Amaranthus hypochondriacus* (Ahypo), *A. cruentus* (Acru), *A. caudatus* (Acau) and *A. hybridus* (Ahyb) plants growing in optimal conditions (Op), or subjected to moderate (M) or severe (S) water deficit stress, or allowed to recover from S, 1 day after normal watering was restored (R). Different letters over the box-and-whisker plots represent statistically significant differences at *p* ≤ 0.05 (Tukey Kramer test; *n* = 9). Box-and-whisker plots show high, low, and median values. The results shown are those obtained from a representative experiment that was repeated in the spring-summer and summer-autumn seasons of 2015, respectively, with similar results.

**Figure 12 plants-08-00239-f012:**
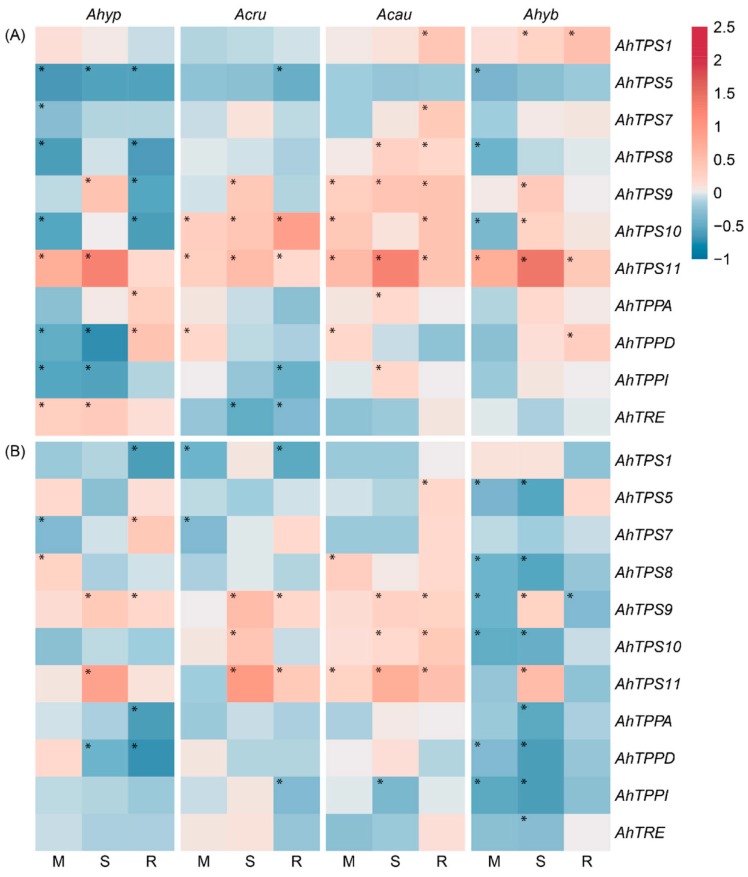
Relative expression of genes involved in trehalose synthesis and degradation in (**A**) leaves and (**B**) roots of *Amaranthus hypochondriacus* (Ahypo), *A. cruentus* (Acru), *A. caudatus* (Acau), and *A. hybridus* (Ahyb) plants subjected to moderate (M) or severe (S) water deficit stress, or allowed to recover from S, 1 day after normal watering was restored (R). Induced (normalized expression values ≥ 1.5; in red) and repressed (normalized expression values ≤ 0.5; in blue) gene expression values, represented in a log_10_ basis, are highlighted with an asterisk at the upper left side of the cells. They were calculated according to the comparative cycle threshold method [28] using the *AhACT7*, *AhEF1a* and *AhβTub5* amaranth genes for data normalization.

**Figure 13 plants-08-00239-f013:**
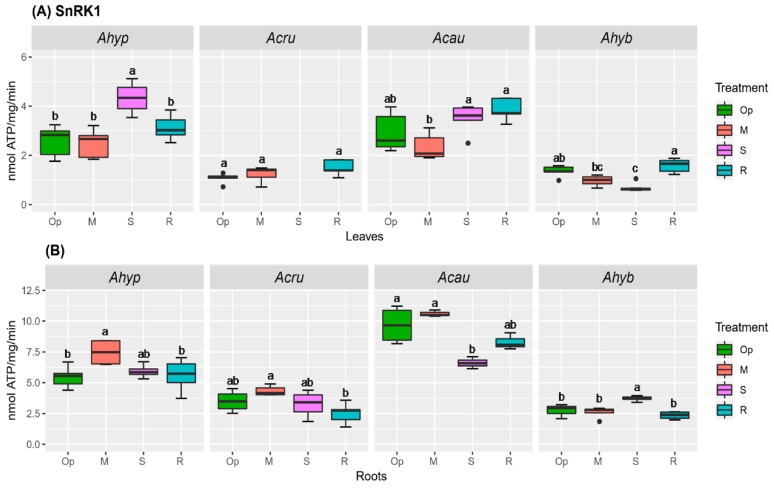
SnRK1 activity in amaranth plants subjected to water-deficit stress. SnRK1 activity in (**A**) leaves and (**B**) roots of *Amaranthus hypochondriacus* (Ahypo), *A. cruentus* (Acru), *A. caudatus* (Acau), and *A. hybridus* (Ahyb) of plants growing in optimal conditions (Op), subjected to moderate (M) or severe (S) water deficit stress, or allowed to recover from S, 1 day after normal watering was restored (R). Different letters over the box-and-whisker plots represent statistically significant differences at *p* ≤ 0.05 (Tukey Kramer test; *n* = 9). Box-and-whisker plots show high, low, and median values. The results shown are those obtained from a representative experiment that was repeated in the spring-summer and summer-autumn seasons of 2015, respectively, with similar results.

**Figure 14 plants-08-00239-f014:**
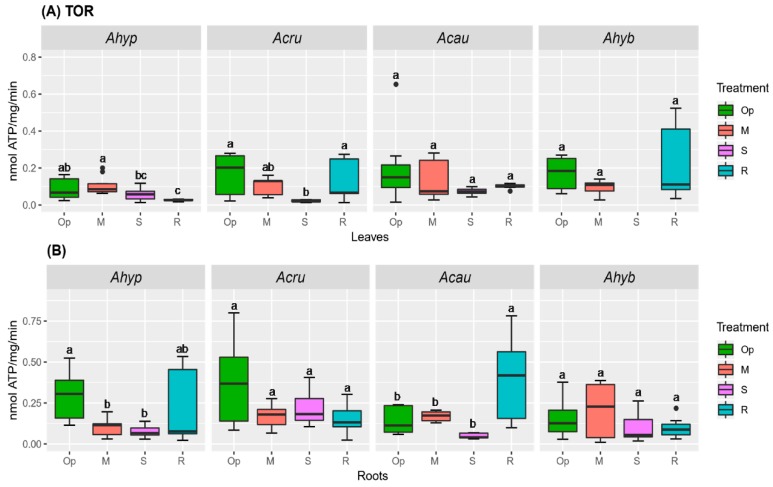
TOR activity in amaranth plants subjected to water-deficit stress. TOR activity in (**A**) leaves and (**B**) roots of *Amaranthus hypochondriacus* (Ahypo), *A. cruentus* (Acru), *A. caudatus* (Acau), and *A. hybridus* (Ahyb) in plants growing in optimal conditions (Op), subjected to moderate (M) or severe (S) water deficit stress, or allowed to recover from S, 1 day after normal watering was restored (R). Different letters over the box-and-whisker plots represent statistically significant differences at *p* ≤ 0.05 (Tukey Kramer test; *n* = 9). Box-and-whisker plots show high, low, and median values. The results shown are those obtained from a representative experiment that was repeated in the spring-summer and summer-autumn seasons of 2015, respectively, with similar results.

**Figure 15 plants-08-00239-f015:**
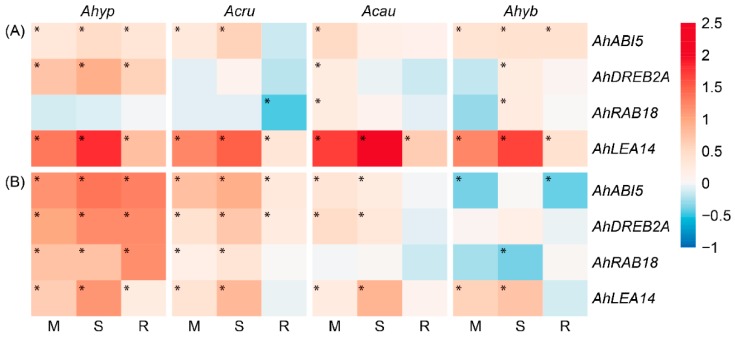
Relative expression of abscisic acid (ABA) marker genes in in (**A**) leaves and (**B**) roots of *Amaranthus hypochondriacus* (Ahypo), *A. cruentus* (Acru), *A. caudatus* (Acau), and *A. hybridus* (Ahyb) plants subjected to moderate (M) or severe (S) water deficit stress, or allowed to recover from S, 1 day after normal watering was restored (R). Induced (normalized expression values ≥ 1.5; in red) and repressed (normalized expression values ≤ 0.5; in blue) gene expression values, represented in a log_10_ basis, are highlighted with an asterisk at the upper left side of the cells. They were calculated according to the comparative cycle threshold method [28] using the *AhACT7*, *AhEF1a* and *AhβTub5* amaranth genes for data normalization.

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
