# Peer review of "Identification of Factors Linked to Higher Water-Deficit Stress Tolerance in Amaranthus hypochondriacus Compared to Other Grain Amaranths and A. hybridus, Their Shared Ancestor"

_plants, 2019, doi:10.3390/plants8070239_

Round 1
Reviewer 1 Report
The manuscript falls within the general scope of the journal. However, the paper can be accepted after major revisions.
The title should be written in a way more related and specific to the obtained results or to the most interesting result.
Introduction is not concluded by a clear hypothesis, which will be tested and discussed.
What kind of randomization was applied for the experiment?
Indicate the PPFD to which the plants were exposed. What about air humidity? What was the air temperature? What constituted a replicate? Were measurements based on one leaf/root, one group of leaves/roots?
Could you clearly indicate what biological and technical replicates are in you research?
It is unclear how plant samples were collected for analyses, whole leaves, whole seedlings, leaf discs?
The physiological state can change along one simple leaf. Did the author use the same leaf part and the leaves of the same sizes for measurements?
The extent of water stress was not quantified (RWC, leaf water potential). This may be absolutely necessary to establish whether (i) the level of leaf dehydration was the same for all species, and (ii) leaf dehydration withstood by each species was equal or not. Do the authors are sure that the plant water status was the same for studied species?
The discussion needs to also include the limitations of this study.
Reviewer 2 Report
The paper “Identification of Factors Responsible for Differential Water-Deficit Stress Tolerance in Grain Amaranths and Their Shared Ancestor Amaranthus hybridus” reports a wide characterization of water deficit stress response in four different Amaranthus species with the aim to elucidate drought tolerance mechanisms that operate in grain amaranth. The characterization is mainly at metabolomics level with the analysis of compounds involved in osmotic adjustment, sugar starvation response and at transcriptomic level with the analysis of gene involved in ABA network.
Overall, the results reported are interesting and the experiments are well conducted. The authors performed a very deep analysis both on leaf and on roots of plants subjected to moderate and severe water deficit raffinose family oligosaccharides deficit tolerance level is interesting and may allow highlighting different mechanisms of response.
Given the complexity of the response to abiotic stress, which has been considered by the authors both in the introduction and in the discussion, the authors decided to study in details some of the key mechanisms of water-deficit response. They combined biochemical, metabolomics and transcriptomic analyses to test the synthesis and degradation of several non-structural carbohydrates, raffinose family oligosaccharides, amino acids, and the regulation of some abscisic acid (ABA) marker genes.
Specific comments
Some general suggestions about Figures and Tables
The figures from 1 to 12 report the data about the four species (leaves and roots) in response to four treatments (Op, M, S, R).
The figures could be standardized since there are different types of graphs, which can be confusing. In particular, Fig. 1 and 2 with lines, all the other with bars; some are grouped according to species, others according to treatments, and so on. I think it would be better to choose a uniform format.
The tables reporting gene expression analysis could be merged in a heat map for example.
Introduction
Line 65 and line 70, avoid repetitions.
Results
The comments to Figures are not complete or are not even correct in several cases. I suggest an overall revision.
In particular, I want to underline the following:
Line 125 ‘The high Hex/Suc ratio….’ this parameter has not been shown in any figure or tables, so it is necessary to explain better within the text.
Line 134-136 ‘Amylolytic activity was almost uniformly induced by WDS and R in leaves of all treated plants (Figure 6A)’.
This is not correct since only in Ahypo the amylolytic activity is induced in R, in all the other species it is similar or even lower than in mild stress treatment.
Line 143
‘The most relevant results…’ in this phrase something is missing, it is not clear.
Line 322
The description of the expression of AhTPS9 is not correct since it is not ‘ induced by both SWDS and R in all species tested…’, more correctly it is induced by SWDS only in Ahypo and Acru, in accordance to what stated in the description of the table (line 341)‘ Induced (normalized expression values ≥ 2.0; …)‘
In all Tables reporting gene expression data, it is not clear why some values lower than 2 are in bold if you consider a gene induced when normalized expression values are ≥ 2.0. Please, correct.
Line 336-339 please consider a revision of this phrase, since is a bit confusing
Line 354 Figure 12 A is cited, while not all the other panels are, why? Please revise.
Paragraph 2.6 Please consider revising the whole description, which is not clear and in some cases not even correct.
Line 368 ‘…AhDREB2C transcription factor-encoding gene was ubiquitously induced…’ what do you mean with ubiquitously?
Line 371 ‘…the ample expression of the AhABI5 gene…’ considering the data in Tab 5 and 6 I wouldn’t say that AhABI5 has an ample expression in leaves
Line 376-378 you say ‘All the above ABA-marker genes were more persistent and strongly induced in roots of WDS-tolerant plants, particularly in A. hypochondriacus (Table 6).’ What do you mean with more persistent?
As I suggested before, this data could be presented as a single heat-map (root and leaves together), allowing an easier reading.
Line 516-519 please consider a revision of this phrase, since is a bit confusing.
Materials and methods
The description of some methods is not exhaustive, but the authors refer to previous papers; I think this is not enough. In particular, an improvement of paragraph 4.4, 4.6, 4.7 is needed.
Conclusions
Line 643 -648
‘Published transcriptomic data revealed that several genes belonging to various categories associated with stress responses showed modified expression levels during WDS…..’
I think this is a sort of discussion which is interesting and should be explained in more detail. They are taken from a previous work of the same authors, in which only Ahypo is in common with this presented work. This data could be discussed considering, for example, the same genes whose expression was tested by qPCR in the present work to confirm eventually the same kind of regulation in response to stress (if the stress is performed in the same way).
In Table S3 some rows are highlighted in yellow, why?
Reviewer 3 Report
Recommendation: Accepted with major modifications
In this manuscript, Gonzalez-Rodriguez and co-workers presented a biochemical and molecular characterization of the response to water deficit in four Amaranthus species, three grain amaranths and their putative ancestor. In particular, the author analysed and discussed the accumulation of different osmolytes and sugars, the activity of enzymes involved in their biosynthesis and/or catabolism and the expression level of genes involved in signalling of stress response. In the discussion, the authors principally correlated these data to the different levels of tolerance observed in these four Amaranthus species under water deficit. A consistent amount of data is presented and the manuscript resulted interesting and publishable. However, some concerns are present and some modifications are required before a publication in Plants.
The most critical point is the absence of phenotypic or physiological data about the different drought response of the four Amaranthus species. All the data discussion is based on these differences, since the metabolite accumulation levels are correlated with the tolerance level of the Amaranthus species, with A. hypocondriacus declared as the most tolerant. Then, the presentation of data showing the levels of drought tolerance in paragraph 2.1 is needed: a photographical documentation about the plant phenotype (whole plant, leaf wilting etc.) and/or physiological data (for instance, the relative water content of leaves) in the considered conditions (Optimal, MWDS, SWDS, Recovery), and/or the number of plants that recovered after stress, correlated with a statistical analysis (a single percentage is not sufficient). It is not possible to consider the manuscript for publication without convincing data about the declared tolerance level of the Amaranthus species.
Some caution has to be applied when interpreting the data. In particular, the author analysed the expression of some genes involved in ABA signalling. Among them, a DREB gene was present. this gene was named as AhDREB2C (line 368 and TableS2), or AhDREB2A in other parts of the text and it is not clear if it is a 2C or a 2A type. DREB genes belong to a multigenic family and are widely recognized as the principal regulators of the ABA-independent stress response pathway in plants (see the several reviews proposed by Shinozaki and co-workers for further details). The authors referred to the reference n. 65 to justify the inclusion of this DREB gene in the ABA signalling network. However, the role of DREB2C in the cross-talk between ABA -dependent and -independent pathway has been demonstrated by Lee et al. 2010 only in Arabidopsis. The assertion that this AhDREB gene (if it is a 2C and not a 2A) can play a prominent role in ABA-dependent signalling pathway is too speculative. This gene must be excluded from the analysis of ABA signalling pathway. In general, all the discussion about expression data on genes putatively involved in ABA signalling (lines 484-513) is too speculative and must be reorganized.
Moreover, the Conclusions are unclear and confused. The authors presented a list of genes which are obtained from previously published transcriptomic data and whose expression level changes during drought stress. They hypothesised that these genes may be part of a gene network responsible of the A. hypocondriacus tolerance. This hypothesis is based only on expression changes and it is too speculative. Moreover, it is not clear the link between the data presented in the paper and this list of genes. I recommend to rewrite the conclusions, limiting them to the data here reported, and to erase the Table S3, which cannot be presented in this way and do not ameliorate the manuscript.
A further criticism of the manuscript is the organization of data presentation. The fact that the amount of submitted data is consistent requires a higher level of organization of the manuscript. i) In Results, the authors described the data in this order: sugars, RFOs, Proline and Threalose, SnRK1/TOR pathway, ABA signalling. Differently, in Discussion they presented first osmotic adjustment, then ABA response and finally sugar starvation response. I suggest to maintain the same order of the presented data in the sections Results and Discussion. ii) The bar charts should be formatted in order to facilitate the comparison among species, which is the principal aim of this work. In particular, the bar charts of Figs. 7, 8, 11, 12, S1 and S2 should be presented in the same way of the bar charts of Figs 3-6, 9 and 10, where the comparison among species is more simple. iii) When possible, the upper limit used in the vertical axis should be the same in the graphs inserted in the same figure, to facilitate the comparison among data. iv) I suggest to strongly limit the use of abbreviations that make the reading complex. For instance, the use of the acronym DP for “degree of polymerization” is useless, since this definition is present only two times throughout the text. Moreover, the abbreviation used for Water Deficit Stress can be simplify from WDS to WD (or WS) to not weigh the text down.
Further criticisms are present in this manuscript, as reported below.
Introduction
- In recent years, several comparative analyses on varieties showing contrasting phenotypes (tolerance/sensitiveness) under water stress have been performed in many important crops, especially in rice. I recommend to insert few of these papers on rice as further examples.
- At line 70, the authors briefly introduced this work listing the analyses performed. I suggest to introduce the work differently, explaining the meaning of these analysis in the context of the plant response to water stress.
Results
- The abbreviation of Recovery (R) must be presented at line 87 (“with ca. 60% recovery”) and not at line 98.
- I suggest to not write “WDS-tolerant” every time that A. hypocondricus is named. This weighs the reading down.
- At line 128 the data are referred to Figure 2 and 4, not to Figure 1 and 4.
- The lines 138-141 and 222-227 should be moved in the Discussion.
- At line 357, the data are referred to Tables S1a and S1b, not S2a and S2b.
Material and Methods
- In the subparagraph “4.3. Extraction of total RNA and gene expression analysis by RT-qPCR” the method for RNA extraction was not described.
Supplementary Materials description
- The numbering of supplemental tables is incorrect.
Round 2
Reviewer 1 Report
The manuscript was revised according to comments that were addressed to previous version of the paper. Therefore, in current form, the work is ready for publication in Plants..
Reviewer 3 Report
In the revised manuscript, all of the requested modifications have been accepted by the authors, consistent with the modifications required by Reviewer 1.
Just two minor modifications are required in the new version:
- figure 9 overlaps part of the capture text. Some lines must be inserted to avoid it;
- in the new text insertions, the authors often use the Saxon Genitive, which is not adequate to scientific language.
Overall this manuscript is improved compared to the previous version; whenever these last modifications will be made, the paper will be publishable in Plants.
